 

ⓐ | **Open Peer Review** | Clinical Microbiology | Research Article

# Rapid molecular detection of respiratory pathogens in patients admitted with suspected community-acquired pneumonia: secondary analysis of a randomized controlled trial

Mariana B. Cartuliares,[1,2] Helene Skjøt-Arkil,[1,2] Christian B. Mogensen,[1,2] Steen L. Andersen,[3] Flemming S. Rosenvinge[4,5]

**ABSTRACT** In a randomized trial, we demonstrated that implementing point-of-care (POC) PCR detection of respiratory pathogens in an emergency department (ED) setting did not change overall antibiotic use but led to more targeted treatment and a clinically relevant, albeit non-significant, reduction in length of stay. This study aimed to assess the diagnostic impact of POC-PCR testing in patients with suspected community-acquired pneumonia (CAP) in the ED. This study is a secondary analysis of a Danish multicenter randomized controlled trial (RCT) that included patients aged ≥18 years admitted to the ED between March 2021 and February 2022. In the primary trial, patients were randomly assigned to one of two parallel groups: (i) intervention group: POC-PCR in addition to standard care, or (ii) control group: standard care only (SCO). The present secondary analysis focuses exclusively on patients in the intervention group, where both POC-PCR and standard culture-based diagnostics were performed. The diagnosis of CAP was determined by an expert panel. In the intervention group, 145 patients had a lower respiratory tract (LRT) specimens analyzed using both POC-PCR and culture. POC-PCR identified notably more microorganisms compared with culture (187 vs 34). The agreement between positive results obtained by POC-PCR and culture was 6 of 19 (31.6%) for *Streptococcus pneumoniae* and 16 of 61 (26%) for *Haemophilus influenzae*. Concentrations of ≥$10^7$ copies/mL detected by POC-PCR for *S. pneumoniae* and *H. influenzae* were associated with a diagnosis of CAP. Of the 67 patients identified with *S. pneumoniae* and/or *H. influenzae*, 46 (69%) received targeted treatment, 57 (85%) received adequate treatment, and 10 (15%) received inappropriate treatment. The rapid pathogen detection by POC-PCR is likely to improve diagnostic accuracy and promote appropriate use of antibiotics in CAP patients.

**IMPORTANCE** Our findings suggest that point-of-care (POC)-PCR testing can significantly enhance the diagnostic accuracy for CAP in the emergency department (ED), thereby facilitating more targeted and effective antimicrobial treatment. In light of growing concerns about antimicrobial resistance and the pressing need for rapid diagnostics in acute care settings, we believe our study holds considerable relevance for clinical practice in the ED.

**KEYWORDS** point of care, antibiotics, PCR, emergency department

Address correspondence to Mariana B. Cartuliares, mbc@rsyd.dk.

The authors declare no conflict of interest.

See the funding table on p. 10.

Community-acquired pneumonia (CAP) is one of the most common infections in emergency departments (EDs) and one of the leading causes of death worldwide (1, 2).

Point-of-care (POC) polymerase chain reaction (PCR) panels for the detection of both viral and bacterial respiratory pathogens reduce time to results and increase the number of microbiological findings compared to sputum culture (3–9). This is essential to

facilitate early appropriate treatment and thereby reducing adverse outcomes, lowering mortality rates, and mitigating the emergence of antimicrobial resistance (8–11). However, while most existing studies have been conducted in intensive care unit (ICU) settings, the diagnostic utility and clinical impact of POC-PCR in emergency department (ED) patients with suspected community-acquired pneumonia (CAP) remains less well characterized. In addition, evaluating the impact of recent antimicrobial treatment is essential, as prior studies have shown that approximately one-third of CAP patients receive antibiotics within one month before admission, leading to a fourfold reduction in diagnostic yield (12, 13).

This study is a secondary analysis of a randomized trial involving 379 patients where we evaluated the impact of POC-PCR on antibiotic prescribing patterns (14). Although the primary analysis did not show a reduction in use of broad-spectrum antibiotics, exploratory analyses indicated an association between POC-PCR and the use of targeted antibiotics in a small subset of culture-positive patients, as well as a potential reduction in length of stay (LOS) by approximately 20% (equivalent to one day). From a clinical standpoint, these findings may be of considerable significance and merit further investigation.

Accordingly, this secondary analysis aimed to assess the diagnostic contribution of POC-PCR in CAP patients in an ED setting and has the following objectives:

i. To examine the microbiological agreement between culture and POC-PCR in an ED setting,
ii. To examine the correlation between microbiological findings, pathogen concentrations, and a reference diagnosis of CAP,
iii. To examine the impact of previous antibiotic treatment on POC-PCR and culture results, and
iv. To describe the use of target and adequate antibiotics within 4 h after admission to the ED

## MATERIALS AND METHODS

This study was a secondary analysis of an RCT. For detailed information about the primary trial, we refer to the main study (14) and the published study protocol (15). The trial ran from March 2021 to February 2022 at three Danish EDs: Lillebælt Hospital in Kolding, Hospital Sønderjylland in Aabenraa, and Odense University Hospital in Odense. The processing of data was approved by the Region of Southern Denmark (no. 20/60508) cf. art 30 of The EU General Data Protection Regulation, approved by the Regional Committee on Health Research Ethics for Southern Denmark (S-20200188), registered by ClinicalTrials.gov (NCT04651712), and conducted according to the Declaration of Helsinki Ethical principle for medical research involving human subjects.

### Participants

Adults (≥18 years) admitted to the ED with suspected CAP who could provide verbal and written consent were enrolled in the study if the attending physician identified at least one of the following pulmonary symptoms: dyspnea, cough, expectoration, chest pain, or fever. Patients were excluded if they could not deliver a sputum sample, participation delayed urgent treatment, the patient was transferred to intensive care, the patient had been admitted within the last 14 days, had COVID-19 infection at admission, was pregnant, or had severe immunodeficiency (14). Patients were included consecutively on weekdays from 10 AM to 8 PM.

### Procedure

This study focused exclusively on the intervention arm of the RCT, where CAP patient samples were analyzed using both POC-PCR and culture. Tracheal secretions were collected in accordance with Danish guidelines (16), though expectorated sputum

was accepted when tracheal suction was unsuccessful. Test results, along with a short interpretation guideline (Interpretation Guideline, S1) providing antibiotic treatment recommendations, were promptly delivered to the attending physician upon availability.

## Point-of-care polymerase chain reaction

Without delay, samples were analyzed using the Biofire FilmArray Pneumonia Panel Plus (bioMérieux, Marcy l'Etoile, France), a fully automated, closed multiplex PCR system with a turnaround time of approximately 75 min. The panel detects 18 bacterial pathogens, nine viral pathogens (17) (Table S2). Concentrations of typical colonizing, opportunistic pathogenic bacteria were reported to the nearest whole log as gene copies/mL ranging from $10^4$ to $10^7$ copies/mL.

## Routine culture and routine PCR

All samples were submitted directly for culture at the local Department of Clinical Microbiology. Part of the sputum sample was transferred to a 5% blood agar plate and to a chromogenic and/or selective agar. The inoculum was streaked over the agar surface, and blood agar plates were inoculated with a *Staphylococcus* streak to allow the growth of *H. influenzae*. Plates were incubated at 35°C—blood agar in a 5% CO2 atmosphere, and other plates in normal atmospheric conditions. After 1 to 2 days of incubation, pathogens were identified by matrix-assisted laser desorption/ionization-time of flight mass spectrometry (BioTyper/Bruker [Bruker Daltronics, Bremen, Germany] or Vitek MS [bioMérieux, Marcy-l'Etoile, France]). The optochin susceptibility test was used to differentiate *S. pneumoniae* from other mitis group streptococci. Results were reported semiquantitatively as few, some, or numerous. Samples without growth of pathogens were reported as "no growth of pathogens" or "upper airway microbiota." Routine PCR was performed if requested by the referring physician (e.g., for *Legionella pneumophila* or influenza virus). The four Departments of Clinical Microbiology largely used different PCR setups and analyses.

## Community-acquired pneumonia diagnosis

Two reference diagnoses of CAP were assigned to all patients. The first diagnosis was determined through an expert panel assessment, consisting of one infectious disease specialist and one emergency medicine specialist at each study site.

They had access to all patients' data recorded in the medical records within the first week of admission, including clinical information, laboratory test results (biochemistry and microbiology, including POC-PCR), chest X-ray, and computed tomography. The experts were blinded to each other and independently registered their assessments in a standardized electronic template (18) in the study database. Disagreements were discussed until consensus was reached.

The second reference diagnosis was based on high-resolution computed tomography (HRCT), selected as the gold-standard imaging modality for CAP in alignment with current guidelines (19, 20). The diagnosis of CAP was established based on routine evaluations by a radiology specialist who was blinded to the culture and POC-PCR results.

## Categorization of antibiotic treatment

We categorized antibiotic treatment administered within four hours of admission as targeted, adequate, or inappropriate, based on POC-PCR results and two distinct pathogen groups: The first group included only the primary pathogens associated with CAP, *S. pneumoniae*, and *H. influenzae*, while the second group was expanded to include *S. pneumoniae, H. influenzae, Moraxella catarrhalis, Pseudomonas aeruginosa*, and *Staphylococcus aureus*. We defined targeted antibiotics as antibiotics directed against the bacterial pathogen without unnecessary broad-spectrum coverage, and adequate antibiotics as any antibiotics effective against the detected bacterial pathogen. Antibiotic

treatment was defined as inappropriate if it was inactive against the detected pathogen or not relevant for treating pneumonia (Table S3).

We excluded *Enterobacterales* and *Acinetobacter* from the classification of antibiotic treatment, as these bacteria typically represent colonization in non-hospitalized patients and are less likely to cause CAP (16).

## Statistical analysis

The sample size was determined by the primary randomized controlled trial.

For descriptive analysis, categorical variables, such as bacterial or viral detection, CAP diagnosis (positive or negative), HRCT findings (positive or negative), POC-PCR concentrations (categorized by copies/mL), and antibiotic treatment within 1 month prior to admission are presented as counts (n) and percentages (%).

We used McNemar's test with an alpha level of 0.05 to assess differences between specific POC-PCR and culture findings. Additionally, we (i) calculated the agreement between the two methods, (ii) examined the relationship between the concentrations reported by POC-PCR for specific bacteria and the two CAP reference diagnoses, (iii) graphically illustrated the impact of prior antibiotic treatment on microbiological results from both POC-PCR and culture, and (iv) described the use of targeted, adequate, and inappropriate antibiotic treatments based on the POC-PCR results.

We used STATA 17.0 (StataCorp., TX, USA) for data analysis and Microsoft PowerPoint 2016 (Microsoft Corp., USA) for illustrations.

## RESULTS

We included 145 patients who had lower respiratory tract specimens analyzed by both POC-PCR and culture (Fig. 1).

## Agreement between POC-PCR and culture

Lower respiratory tract samples were collected by tracheal suction (116 samples, 80%) and expectorated sputum (29 samples, 20%). POC-PCR detected 187 bacterial and viral pathogens in 109 (75.2%) of the 145 samples (Table 1). A single potential pathogen was identified in 53 samples (36.6%), while two or more possible pathogens were identified in 56 samples (38.6%). In 36 samples (24.8%), no pathogen was detected.

The overall agreement between pathogens detected by POC-PCR and culture is summarized in Table 2.

For *S. pneumoniae*, 19 samples tested positive by at least one of the methods, but only six (37.5%) were positive by both. The POC-PCR concentrations in these six concordant samples were reported as $10^6$ copies/mL (three samples) and $10^7$ copies/mL (three samples)

Only 16 (26.2%) of the 61 samples positive for *H. influenzae* were detected by both methods. Notably, all 16 culture-positive *H. influenzae* cases were also identified by POC-PCR. Among these, 15 samples had a POC-PCR concentration of $10^7$ copies/mL, while the concentration for the remaining sample was not reported.

Correlation between POC-PCR microorganism concentrations and community-acquired pneumonia diagnosis.

For *S. pneumoniae* and *H. influenzae*, higher POC-PCR concentrations were positively correlated with both the expert and the HRCT CAP diagnosis (Table 3). In contrast, for other microorganisms, there was no evident correlation between POC-PCR concentration and the CAP reference diagnoses.

## The impact of previous antibiotic treatment on microbiological results

In total, 36 (25%) of the 145 patients had received antibiotics within one month prior to admission. Prior antibiotic treatment clearly reduced the detection sensitivity for *S. pneumoniae* in both culture and POC-PCR testing (Fig. 2).

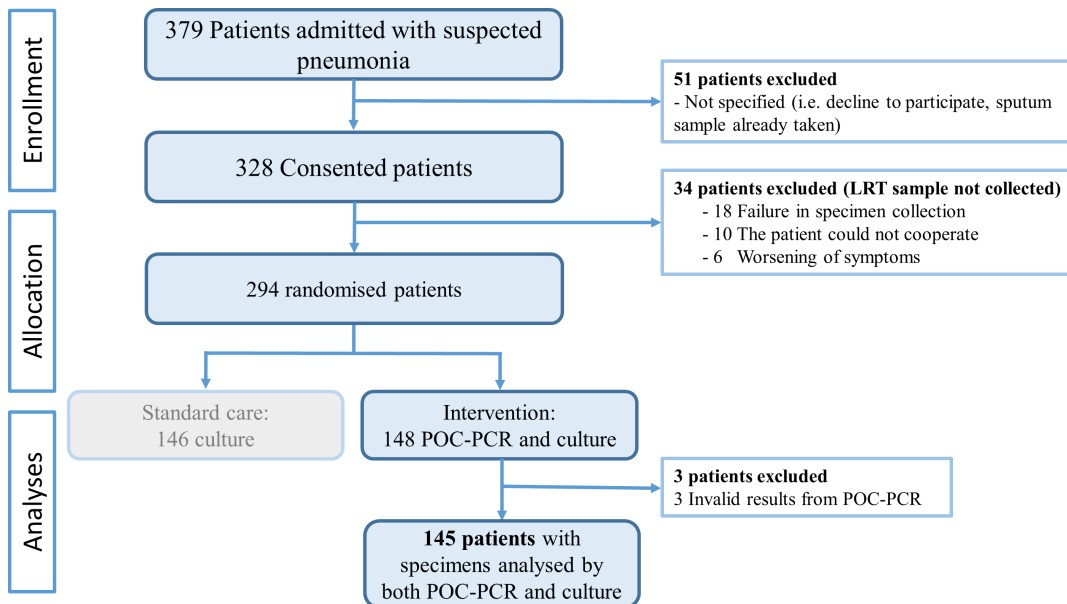

**FIG 1** Dark blue boxes indicate the population in which lower respiratory tract (LRT) specimens were analyzed using both point-of-care polymerase chain reaction (POC-PCR) and culture.

## Use of target and adequate antibiotics within 4 h of admission

Based on the classification in which *S. pneumoniae, H. influenzae, M. catarrhalis, P. aeruginosa,* and *S. aureus* were considered significant pathogens, we identified 90

**TABLE 1** Respiratory pathogens detected by point-of-care (POC) polymerase chain reaction (PCR) and culture from 145 patients admitted with suspected community-acquired pneumonia[e]

| Microbial etiology | Analysis methods (*n* = 145) | | | *P*-value |
|---|---|---|---|---|
| | POC-PCR[a] | Culture | Detected by both POC-PCR and culture | |
| Bacteria | 152 | **31** | | |
| S. pneumoniae | 16 | 9 | 6 | 0.05 |
| H. influenzae | 61 | 16 | 16 | <0.001 |
| M. catarrhalis | 16 | 1 | 1 | <0.001 |
| P. aeruginosa | 3 | 0 | 0 | 0.08 |
| S. aureus | 30 | 2 | 2 | <0.001 |
| Enterobacterales | 19 | 2 | 1 | <0.001 |
| L. pneumophila | 0 | 1 | 0 | 0.31 |
| Others[b] | 7 | –[f] | – | |
| Viruses | **35** | **3** | | |
| Human rhinovirus/enterovirus | 14 | – | – | – |
| Coronavirus (CoV)[c] | 7 | – | – | – |
| Parainfluenza virus | 5 | – | – | – |
| Respiratory syncytial virus | 4 | – | – | – |
| Human metapneumovirus | 3 | – | – | – |
| Influenza A virus | 2 | – | – | – |
| SARS-CoV-2[d] | – | 3 | – | – |

[a]POC-PCR: point-of-care polymerase chain reaction.
[b]Others: *Acinetobacter baumannii* complex and beta-hemolytic streptococci.
[c]Coronavirus variants 229E, OC43, HKU1 and NL63.
[d]SARS-CoV-2 were not included in the POC-PCR. No findings of *C. pneumoniae, M. pneumoniae, S. pyogenes,* influenza B virus, MERS-CoV.
[e]McNemar's test was used to assess differences between specific POC-PCR and culture findings.
[f]–, data not available (no information was recorded).

**TABLE 2** Overall agreement between POC-PCR and culture

| Findings | POC-PCR (negative) | POC-PCR (positive) | Total |
|---|---|---|---|
| POC-PCR *S. pneumoniae*, n (%) | 129 (89.0) | 16 (11.0) | 145 (100.0) |
| Culture *S. pneumoniae*, n (%) | | | |
| Negative | 126 (97.7) | 10 (62.5) | 136 (93.8) |
| Positive | 3 (2.3) | 6 (37.5) | 9 (6.2) |
| Overall agreement *S. pneumoniae* | | | 91.0% |
| POC-PCR *H. influenzae*, n (%) | 84 (57.9) | 61 (42.1) | 145 (100.0) |
| Culture *H. influenzae*, n (%) | | | |
| Negative | 84 (100.0) | 45 (73.8) | 129 (89.0) |
| Positive | 0 (0.0) | 16 (26.2) | 16 (11.0) |
| Overall agreement *H. influenzae* | | | 69.0% |
| POC-PCR *M. catarrhalis*, n (%) | 129 (89.0) | 16 (11.0) | 145 (100.0) |
| Culture *M. catarrhalis*, n (%) | | | |
| Negative | 129 (100.0) | 15 (93.8) | 144 (99.3) |
| Positive | 0 (0.0) | 1 (6.3) | 1 (0.7) |
| Overall agreement *M. catarrhalis* | | | 89.7% |
| POC-PCR *P. aeruginosa*, n (%) | 142 (97.9) | 3 (2.1) | 145 (100.0) |
| Culture *P. aeruginosa*, n (%) | | | |
| Negative | 142 (100.0) | 3 (100.0) | 145 (100.0) |
| Positive | 0 (0.0) | 0 (0.0) | 0 (0.0) |
| Overall agreement *P. aeruginosa* | | | 97.9% |
| POC-PCR *S. aureus*, n (%) | 115 (79.3) | 30 (20.7) | 145 (100.0) |
| Culture *S. aureus*, n (%) | | | |
| Negative | 115 (100.0) | 28 (93.3) | 143 (98.6) |
| Positive | 0 (0.0) | 2 (6.7) | 2 (1.4) |
| Overall agreement *S. aureus* | | | 81.7% |
| POC-PCR *Enterobacterales*, n (%) | 126 (86.9) | 19 (13.1) | 145 (100.0) |
| Culture *Enterobacterales*, n (%) | | | |
| Negative | 125 (99.2) | 18 (94.7) | 143 (98.6) |
| Positive | 1 (0.8) | 1 (5.3) | 2 (1.4) |
| Overall agreement *Enterobacterales* | | | 86.9% |

patients. Of these, 40 patients (44%) received targeted antibiotic treatment, 60 patients (67%) received adequate treatment, and 30 patients (33%) were treated inappropriately. When the classification was restricted to *S. pneumoniae* and *H. influenzae* alone, the total number of patients was 67. Among them, 46 patients (69%) received targeted treatment, 57 patients (85%) received adequate treatment, and 10 patients (15%) were treated inappropriately. Of the 78 patients without *S. pneumoniae* or *H. influenzae*, 60 (77%) received no antibiotics or appropriate empirical treatment according to our CAP guideline. Of the 55 patients without a significant pathogen based on the expanded definition, 47 (85%) either did not receive antibiotics or were administered appropriate empirical treatment in accordance with our CAP guideline.

The complete list of pathogens with their respective classifications is presented in Table S4.

## DISCUSSION

### Key findings

This secondary analysis of a randomized controlled trial examined POC-PCR and culture results from 145 patients. POC-PCR identified significantly more microorganisms, most notably detecting nearly twice as many *S. pneumoniae* and four times more *H. influenzae* compared to culture. Overall, the agreement between positive POC-PCR and positive culture results was low. POC-PCR concentrations greater than $10^6$ copies/mL

**TABLE 3** Correlation between pathogens detected by culture and POC-PCR (including concentration) and reference CAP diagnosis (expert diagnosis and high-resolution computed tomography)[c]

| Outcome | CAP[a] negative | CAP[a] positive | HRCT[b] negative | HRCT[b] positive |
|---|---|---|---|---|
| Total n (%) | 56 (38.6) | 89 (61.4) | 66 (45.5) | 79 (54.5) |
| *S. pneumoniae* | | | | |
| Culture | 1 (1.8) | 8 (9.0) | 5 (7.6) | 4 (5.1) |
| POC-PCR | 1 (1.8) | 15 (16.9) | 5 (7.6) | 11 (13.9) |
| $10^4$ copies/mL | 0 (0.0) | 2 (2.2) | 0 (0.0) | 2 (2.5) |
| $10^5$ and $10^6$ copies/mL | 1 (1.8) | 8 (8.9) | 3 (4.5) | 6 (7.6) |
| $10^7$ copies/mL | 0 (0.0) | 5 (5.6) | 2 (3.0) | 3 (3.8) |
| *H. influenzae* | | | | |
| Culture | 2 (3.6) | 14 (15.7) | 4 (6.1) | 12 (15.2) |
| POC-PCR | 11 (19.6) | **50 (56.2)** | **25 (37.9)** | **36 (45.6)** |
| $10^4$ copies/mL | 3 (5.4) | 5 (5.6) | 4 (6.1) | 4 (5.1) |
| $10^5$ and $10^6$ copies/mL | 5 (8.9) | 13 (14.6) | 12 (18.2) | 6 (7.6) |
| $10^7$ copies/mL | 2 (3.6) | **32 (36.0)** | 8 (12.1) | **26 (32.9)** |
| *M. catarrhalis* | | | | |
| Culture | 0 (0.0) | 1 (1.1) | 0 (0.0) | 1 (1.3) |
| POC-PCR | 5 (8.9) | 11 (12.4) | 4 (6.1) | 12 (15.2) |
| $10^4$ copies/mL | 1 (1.8) | 1 (1.1) | 1 (1.5) | 1 (1.3) |
| $10^5$ and $10^6$ copies/mL | 3 (5.4) | 8 (9.0) | 3 (4.5) | 8 (9.0) |
| $10^7$ copies/mL | 1 (1.8) | 2 (2.2) | 0 (0.0) | 3 (3.8) |
| *P. aeruginosa* | | | | |
| Culture | 0 (.) | 0 (.) | 0 (.) | 0 (.) |
| POC-PCR | 2 (3.6) | 1 (1.1) | 2 (3.0) | 1 (1.3) |
| $10^4$ copies/mL | 1 (1.8) | 0 (0.0) | 1 (1.5) | 0 (0.0) |
| $10^5$ and $10^6$ copies/mL | 1 (1.8) | 1 (1.1) | 1 (1.5) | 1 (1.3) |
| $10^7$ copies/mL | 0 (.) | 0 (.) | 0 (.) | 0 (.) |
| *S. aureus* | | | | |
| Routine culture | 1 (1.8) | 1 (1.1) | 0 (0.0) | 2 (2.5) |
| POC-PCR | **12 (21.4)** | **18 (20.2)** | 13 (19.7) | **17 (21.5)** |
| $10^4$ copies/mL | 6 (10.7) | 9 (10.1) | 8 (12.1) | 7 (8.9) |
| $10^5$ and $10^6$ copies/mL | 4 (7.2) | 7 (7.9) | 5 (7.6) | 6 (7.6) |
| $10^7$ copies/mL | 2 (3.6) | 1 (1.1) | 0 (0.0) | 3 (3.8) |
| *Enterobacterales* | | | | |
| Routine culture | 2 (3.6) | 0 (0.0) | 1 (1.5) | 1 (1.3) |
| POC-PCR | **13 (23.2)** | 6 (6.7) | 9 (13.6) | 10 (12.7) |
| $10^4$ copies/mL | 8 (14.3) | 5 (5.6) | 6 (9.1) | 7 (8.9) |
| $10^5$ and $10^6$ copies/mL | 6 (10.8) | 3 (3.3) | 6 (9.0) | 3 (3.8) |
| $10^7$ copies/mL | 0 (.) | 0 (.) | 0 (.) | 0 (.) |

[a]CAP: community-acquired pneumonia determined by expert panel.
[b]HRCT: high resolution computed tomography.
[c]Findings with >20% positivity are shown in bold.

for *S. pneumoniae* and *H. influenzae* were strongly associated with the reference CAP diagnoses, while no clear association was observed for other pathogens. Prior antibiotic treatment notably reduced the detection of *S. pneumoniae* by both culture and POC-PCR, but had a lower effect on other pathogens. Almost 3/4 of the patients identified with *S. pneumoniae* and/or *H. influenzae* received a targeted treatment, and 85% received adequate antibiotic treatment within 4 h of admission. These proportions decreased when including other pathogens.

## Comparison to previous studies

In the primary randomized trial, we did not demonstrate a significant increase in the relative use of no or narrow-spectrum antibiotics (the primary outcome) (14). However,

| Microorganisms | Culture (145) | | POC-PCR (145) | |
|---|---|---|---|---|
| | No antibiotics (109, 75%) | Antibiotics (36, 25%) | No antibiotics (109, 75%) | Antibiotics (36, 25%) |
| *S. pneumoniae* | 9 (8 %) | 0 | 15 (13 %) | 1 (3 %) |
| *H. influenzae* | 13 (12 %) | 3 (8 %) | 47 (43 %) | 14 (39 %) |
| *M. catarrhalis* | 0 | 1 (3%) | 13 (12%) | 3 (8 %) |
| *P. aeruginosa* | 0 | 0 | 2 (2%) | 1 (3%) |
| *S. aureus* | 2 (2 %) | 0 | 24 (22 %) | 6 (17 %) |
| Enterobacterales | 1 (1 %) | 1 (3%) | 13 (12%) | 6 (17 %) |

FIG 2 Findings of microorganisms presented in numbers (N) and percentage (%) from patients who were treated and not treated with antibiotics within one month prior to admission. Gray color: microorganism detected. Blue color: microorganism not detected.

we observed a nearly 20% reduction in length of stay (from 5.2 to 4.2 days), which, although not statistically significant, is clinically noteworthy (14). We hypothesized that this reduction might be attributed to a more accurate, diagnostics-driven approach to antibiotic use—essentially, administering the right antibiotic to the right patient. This was supported by a logistic regression analysis, which showed a significantly increased use of targeted ($P = 0.008$) and adequate ($P = 0.001$) antibiotics in the POC-PCR arm within 48 h of admission, observed in a small subset of culture-positive patients (55 out of 291 patients, including 26 of 145 in the POC-PCR arm) (14). Within four hours, 15 of 26 patients (57.7%) received targeted antibiotics, and 19 of 26 patients (73.1%) received adequate antibiotics. These numbers are comparable to those in this secondary analysis, expanding the observation from culture results to POC-PCR findings and from 26 to 145 patients. This further supports the hypothesis that POC-PCR may influence individual treatment decisions.

Both the increased detection of airway pathogens by POC-PCR compared with culture and the high overall agreement (>80%) between the two methods are consistent with previous studies (8, 21, 22). However, the high concordance may largely reflect agreement on negative samples, particularly for pathogens with relatively low prevalence (23). Notably, for the most prevalent bacterium, *H. influenzae*, the overall agreement was only 69% in our study. The ability of POC-PCR to detect viral respiratory pathogens is important, as it may support withholding antibiotic therapy (6, 7, 24). Compared with previous studies, we detected relatively few viral pathogens, likely due to the impact of SARS-CoV-2 containment measures during the study period (5, 25). It is reasonable to assume that a higher number of viral pathogens would have been identified in a non-pandemic setting (26).

## Clinical implications

In this study, we detected nearly four times more *H. influenzae* than *S. pneumoniae*, contrasting the traditional view of *S. pneumoniae* as the leading bacterial cause of CAP. This observation aligns with recent findings from Denmark and Norway, suggesting that *H. influenzae* may now be the predominant bacterial pathogen in CAP (25, 27). The reasons for this shift are unclear but could involve both diagnostic factors (e.g., low sensitivity of culture and increased use of PCR) and epidemiological factors (e.g., a higher prevalence of patients with chronic lung diseases and increased use of pneumococcal vaccines). This trend may challenge current Danish guidelines recommending narrow-spectrum antibiotics, as over 25% of *H. influenzae* isolates are now ampicillin-resistant—most of them beta-lactamase-negative (28). Rapid diagnostics, such as POC-PCR, may serve as a key tool to address this challenge by shifting from a guideline-based to a diagnostics-driven approach—enabling safe, targeted, and judicious use of antibiotics despite evolving CAP etiology.

The high prevalence of *H. influenzae* could be partly due to frequent colonization in certain patient populations, particularly those with chronic obstructive pulmonary disease (COPD), combined with the high sensitivity of POC-PCR (29). However, the clear association between *H. influenzae* (including its reported concentration) and both the CAP-reference diagnosis and the HRCT-pneumonia diagnosis challenges this explanation.

It is noteworthy that 3 (2.3%) of the *S. pneumoniae* isolates were identified as significant pathogens by culture but not detected by POC-PCR. The reason for this discrepancy is unclear, but potential explanations may include lower sensitivity of POC-PCR for certain pneumococcal strains or the possibility that culture failed to differentiate *S. pneumoniae* from other *Streptococcus mitis* group members. Unfortunately, study isolates were not stored, preventing further analysis.

Our study suggests a strong association between the concentrations of *S. pneumoniae* and *H. influenzae* detected by POC-PCR and both the CAP reference diagnosis and HRCT-pneumonia diagnosis. To our knowledge, this association has not been previously investigated in an ED setting. This finding indicates that POC-PCR bacterial concentrations could potentially serve as a marker for diagnosing CAP, though further research is needed to evaluate the diagnostic accuracy of these concentrations for common CAP pathogens in EDs.

Interestingly, the prevalence and concentration of *M. catarrhalis*, *S. aureus*, and *Enterobacterales*, although detected by POC-PCR in 65 of 145 samples, appeared unrelated to both the expert CAP diagnosis and HRCT findings. Since these bacteria are common commensals in the upper airway, the diagnostic significance of these findings should be interpreted with caution. When we applied the broad definition of respiratory pathogenic bacteria, some patients received targeted or adequate treatment for detected *S. pneumoniae* or *H. influenzae* (Table S4), yet their treatment was deemed inappropriate due to co-infection or colonization with another pathogen. In reality, this may not reflect incorrect treatment but rather a clinical prioritization of the most likely pathogens, while disregarding common commensals, such as *M. catarrhalis* and *S. aureus* —in line with the provided interpretation guideline. In this regard, it would be important to investigate whether the concentration of different pathogens detected by PCR can help distinguish true pathogens from commensals. In addition, this crucial interpretive aspect suggests that combining POC-PCR with interpretive guidelines is important to enhance the clinical value in an ED setting.

## Strengths and limitations

A key strength of this study is its multicenter design in a real-world setting, which enhances the generalizability of the results to clinical ED settings. Additionally, we included a reference CAP diagnosis based on an expert panel's assessment of the patient's medical record, including all imaging and laboratory results. However, the study

also has limitations. As a descriptive secondary analysis, the sample size was determined by the primary randomized study, so the findings should be interpreted cautiously and considered hypothesis generating. Moreover, we only included patients who were able to provide consent, and enrollment was restricted to weekdays and daytime hours, potentially leading to a selection bias that may have excluded more severe CAP cases. Previous research has shown higher intensive care admission rates and mortality among patients hospitalized during off-hours (30). Another limitation is that we included only patients suspected of having CAP, which might have led to missing cases with subtle or atypical presentations. This could particularly impact older patients, who often present with ambiguous or non-respiratory symptoms (31).

In conclusion, this study indicates that POC-PCR adds value to the diagnosis of CAP, particularly when factoring in the concentrations of identified etiological bacteria. POC-PCR may facilitate more targeted and adequate treatment and may thereby improve patient outcomes. However, utilizing PCR for the management and diagnosis of CAP presents interpretive challenges that necessitate support from expert guidance.

## AUTHOR AFFILIATIONS

[1]Department of Emergency Medicine, University Hospital of Southern Denmark, Aabenraa, Denmark

[2]Department of Regional Health Research, University of Southern Denmark, Aabenraa, Denmark

[3]Department of Clinical Microbiology, University Hospital of Southern Denmark, Aabenraa, Denmark

[4]Department of Clinical Microbiology, Odense University Hospital, Odense, Denmark

[5]Research Unit of Clinical Microbiology, University of Southern Denmark, Odense, Denmark

## AUTHOR ORCIDs

Mariana B. Cartuliares (iD) http://orcid.org/0000-0003-0923-6960

## FUNDING

| Funder | Grant(s) | Author(s) |
| --- | --- | --- |
| Region of Southern Denmark | A583 | Mariana B. Cartuliares |
| University of Southern Denmark | 17/10636 | Mariana B. Cartuliares |
| | | Helene Skjøt-Arkil |
| | | Christian B. Mogensen |
| Hospital Sønderjylland | 20/20505 | Mariana B. Cartuliares |
| | | Helene Skjøt-Arkil |
| | | Christian B. Mogensen |

## AUTHOR CONTRIBUTIONS

Mariana B. Cartuliares, Conceptualization, Data curation, Formal analysis, Funding acquisition, Investigation, Methodology, Project administration, Writing – original draft, Writing – review and editing | Helene Skjøt-Arkil, Conceptualization, Funding acquisition, Methodology, Project administration, Supervision, Visualization, Writing – review and editing | Christian B. Mogensen, Conceptualization, Funding acquisition, Investigation, Methodology, Resources, Writing – review and editing | Steen L. Andersen, Conceptualization, Validation, Writing – review and editing | Flemming S. Rosenvinge, Conceptualization, Investigation, Methodology, Supervision, Validation, Visualization, Writing – review and editing

## ADDITIONAL FILES

The following material is available online.

## Supplemental Material

**Tables S1 to S4 (Spectrum01260-25-s0001.docx).** Table S1: Interpretation Guideline. Table S2: Targets of the Biofire FilmArray Pneumonia Panel plus. Table S3: Classification of "targeted, adequate and inappropriate" treatment. Table S4: Classification of antibiotic treatment based on CAP pathogens.

## Open Peer Review

**PEER REVIEW HISTORY (review-history.pdf).** An accounting of the reviewer comments and feedback.

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
