## [Reviewer comments · Microbiology Spectrum]

Microbiology Spectrum

Rapid molecular detection of respiratory pathogens in patients admitted with suspected community-acquired pneumonia: Secondary analysis of a randomized controlled trial

Mariana Cartuliales, Helene Skjøt-Arkil, Christian Mogensen, Steen Andersen, and Flemming Rosenvinge

Corresponding Author(s): Mariana Cartuliales, University Hospital of Southern Denmark

Review Timeline:

Submission Date:	April 23, 2025
Editorial Decision:	June 12, 2025
Revision Received:	July 5, 2025
Accepted:	July 10, 2025

Editor: Mark Pandori

Reviewer(s): The reviewers have opted to remain anonymous.

Transaction Report:

DOI: <https://doi.org/10.1128/spectrum.01260-25>

Re: Spectrum01260-25 (**Rapid molecular detection of respiratory pathogens in patients admitted with suspected community-acquired pneumonia: Secondary analysis of a randomized controlled trial**)

Dear Dr. Mariana Bichuette Cartulieres:

Thank you for the privilege of reviewing your work. Below you will find my comments, instructions from the Spectrum editorial office, and the reviewer comments.

Revision Guidelines

Sincerely,
Mark Pandori
Editor
Microbiology Spectrum

Reviewer #1 (Comments for the Author):

This is a well-written, easy to follow manuscript. I found that the methods of analysis were appropriate. I found the data on copy number and agreement with other clinical data very interesting. I did struggle a bit with interpreting the data in Table 3 however because there are so many data points to compare.

Reviewer #2 (Comments for the Author):

The authors conducted a secondary analysis of a randomized trial assessing point-of-care PCR testing in emergency department patients with suspected community-acquired pneumonia, demonstrating that POC-PCR detected more pathogens than culture and was associated with improved targeted antibiotic treatment. This study is highly important as it supports the use of rapid point-of-care PCR testing to enable early and accurate diagnosis of community-acquired pneumonia, which is essential for timely treatment, ultimately helping to reduce mortality rates.

Major Comments:

- The manuscript would benefit from improved organization, particularly in the Discussion section, which currently feels somewhat lengthy and scattered. A more focused and concise discussion highlighting the key findings, their implications, and clinical relevance would strengthen the overall impact.
- In the Abstract, clearly define the intervention group. Readers unfamiliar with the original trial may find it unclear.
- Acronyms such as RCT and HRCT should be defined at first mention to ensure accessibility for all readers.
- In the Introduction, lines 54-58 include study results, which should be relocated to the Results or Discussion. The introduction should instead focus on background information, such as the clinical challenges in diagnosing CAP and the potential advantages of POC-PCR.

Specific Comments:

- Methods (lines 109-110): Specify which genes were targeted and the primers used.
- For both POC-PCR and MALDI-TOF, include the name and model of the equipment used.
- Table S1 needs substantial revision: it begins with a figure, lacks a title, and includes excessive narrative text alongside what appear to be three embedded tables. The formatting appears to be a copy-paste from bullet points. Consider reorganizing this into clearly labeled, separate tables with concise legends.
- Line 139: Indicate which categorical variables were included in the analysis.
- Table 1: Add the name of the statistical test used for calculating p-values.
- Line 181: Ensure spacing is correct between "S. pneumoniae."
- Species names should be consistently italicized throughout the manuscript. Avoid italicizing "spp." (e.g., *Acinetobacter* spp.).
- Figure 2: Include a key explaining what the blue and grey colors represent.
- Line 215: Report the p-value and specify which statistical test was used.

Review of “Rapid molecular detection of respiratory pathogens in patients admitted with suspected community-acquired pneumonia: Secondary analysis of a randomized control trial”

Authors: Mariana B Cartuliales, Helen Skjot-Arkil, Christian B Mogensen, Steen L Anderson and Flemming S Rosenvinge

Comments:

This is a well-written, easy to follow manuscript. I found that the methods of analysis were appropriate. I found the data on copy number and agreement with other clinical data very interesting. I do not have any major suggestions for revisions on this manuscript and would recommend acceptance.

I did struggle a bit with interpreting the data in Table 3, however, because there is a lot of comparison happening in one table. Perhaps pairing the Neg Culture/Neg HRCT and Pos Culture/Pos HRCT columns would help? I am not sure that the color coding helped me. Another option would be to provide a little more commentary in the narrative to fully explain this table.

Dear reviewers

I am grateful for your very thorough review and constructive comments.

Thank you very much for your time.

Best regards

Mariana Bichuette Cartulieres

Reviewer #1 (Comments for the Author):

This is a well-written, easy to follow manuscript. I found that the methods of analysis were appropriate. I found the data on copy number and agreement with other clinical data very interesting. I did struggle a bit with interpreting the data in Table 3 however because there are so many data points to compare.

ANSWER: Thank you for the positive feedback. We agree that Table 3 presents a substantial amount of data.

Reviewer #2 (Comments for the Author):

The authors conducted a secondary analysis of a randomized trial assessing point-of-care PCR testing in emergency department patients with suspected community-acquired pneumonia, demonstrating that POC-PCR detected more pathogens than culture and was associated with improved targeted antibiotic treatment. This study is highly important as it supports the use of rapid point-of-care PCR testing to enable early and accurate diagnosis of community-acquired pneumonia, which is essential for timely treatment, ultimately helping to reduce mortality rates.

Major Comments:

- The manuscript would benefit from improved organization, particularly in the Discussion section, which currently feels somewhat lengthy and scattered. A more focused and concise discussion highlighting the key findings, their implications, and clinical relevance would strengthen the overall impact.

ANSWER: Thank you for addressing this to strengthen the overall impact. We went through the discussion to make sure that the discussion is more focused and organized as you mentioned above. Let us know if it is not clear.

- In the Abstract, clearly define the intervention group. Readers unfamiliar with the original trial may find it unclear.

ANSWER: Thank you for mentioning that. We have defined the intervention group more clearly: “. In the primary trail, patients were randomly assigned to one of two parallel groups: 1) intervention group: POC-PCR in addition to standard care, or 2) control group: standard care only (SCO). The present secondary analysis focuses exclusively on...”

- Acronyms such as RCT and HRCT should be defined at first mention to ensure accessibility for all readers.

ANSWER: Thank you for comment. The acronym RCT was mentioned in the end of the introduction, but we have now also defined it in the abstract, where it first appears. Regarding HRCT, it is first introduced on page 7, where we define it as “high-resolution computed tomography”. We have reviewed the manuscript carefully and could not find any prior use of the acronym before this point.

- In the Introduction, lines 54-58 include study results, which should be relocated to the Results or Discussion. The introduction should instead focus on background information, such as the clinical challenges in diagnosing CAP and the potential advantages of POC-PCR.

ANSWER: Thank you for your comment. The results mentioned in the introduction are not from the present secondary analysis but originate from the primary randomized trial. We believe these findings are clinically relevant and serve as the main rationale for conducting the current study. They form the foundation for our research objectives. We have revised the text in the introduction to clarify this distinction.

Specific Comments:

- Methods (lines 109-110): Specify which genes were targeted and the primers used.

ANSWER: Standard care PCR analyses were performed across four different Departments of Clinical Microbiology, each employing a variety of diagnostic approaches. These included differing PCR setups (e.g., monoplex vs. multiplex), platforms (e.g. in-house vs. commercial assays), and variable primer and probe designs—even when targeting the same pathogens. Given this heterogeneity, we believe that detailed reporting of primer and probe sequences is beyond the scope of this manuscript and would not meaningfully contribute to the interpretation of our findings. Note added to the manuscript.

- For both POC-PCR and MALDI-TOF, include the name and model of the equipment used.

ANSWER: Thank you for this comment. We have added the name and model for the equipment used for the MALDI-TOF. The POC-PCR is described in page 5 lines 94-99 and table supplementary S2.

- Table S1 needs substantial revision: it begins with a figure, lacks a title, and includes excessive narrative text alongside what appear to be three embedded tables. The formatting appears to be a copy-paste from bullet points. Consider reorganizing this into clearly labeled, separate tables with concise legends.

ANSWER: Thank you for pointing this out. You are correct that S1 is not a table, but rather the interpretation guideline provided to clinicians along with the POC-PCR results. We have renamed it “Interpretation Guideline” and updated the text and supplementary materials accordingly.

- Line 139: Indicate which categorical variables were included in the analysis.

ANSWER: Thank you for this comment. We have now added the categorical variables that were investigated in the study.

- Table 1: Add the name of the statistical test used for calculating p-values.

ANSWER: Thank you for mentioning that. We have added the test for Table 1 in the legend “McNemar’s test was used to assess differences between specific POC-PCR and culture findings.”

- Line 181: Ensure spacing is correct between "S. pneumoniae."

ANSWER: Thank you for notice that. We have corrected the space between "*S. pneumoniae*"

- Species names should be consistently italicized throughout the manuscript. Avoid italicizing "spp." (e.g., *Acinetobacter* spp.).

ANSWER: Thank you for pointing this out. We have removed 'spp.' and retained the genus name in italics, in accordance with standard scientific nomenclature

- Figure 2: Include a key explaining what the blue and grey colors represent.

ANSWER: Thank you for your comment. We have added the following explanation to the figure legend:
"Grey: microorganism detected; Blue: microorganism not detected."

- Line 215: Report the p-value and specify which statistical test was used.

ANSWER: Thank you for this comment we have added the p-values and that this results were based on a logistic regression analysis: *"This was supported by a logistic regression analysis which showed a significantly increased use of targeted (p=0.008) and adequate antibiotics (p=0.001) in the POC-PCR arm within 48 hours of admission"*

Re: Spectrum01260-25R1 (**Rapid molecular detection of respiratory pathogens in patients admitted with suspected community-acquired pneumonia: Secondary analysis of a randomized controlled trial**)

Dear Dr. Mariana Bichuette Cartulieres:

Your manuscript has been accepted, and I am forwarding it to the ASM production staff for publication. Your paper will first be checked to make sure all elements meet the technical requirements. ASM staff will contact you if anything needs to be revised before copyediting and production can begin. Otherwise, you will be notified when your proofs are ready to be viewed.

Sincerely,
Mark Pandori
Editor
Microbiology Spectrum